# Development of a Multiplex Droplet Digital PCR Assay for the Detection of *Babesia*, *Bartonella*, and *Borrelia* Species

**DOI:** 10.3390/pathogens10111462

**Published:** 2021-11-11

**Authors:** Ricardo Maggi, Edward B. Breitschwerdt, Barbara Qurollo, Jennifer C. Miller

**Affiliations:** 1Department of Clinical Sciences, The Comparative Medicine Institute, College of Veterinary Medicine, North Carolina State University, 1060 William Moore Drive, Raleigh, NC 27607, USA; ebbreits@ncsu.edu (E.B.B.); baquroll@ncsu.edu (B.Q.); 2Galaxy Diagnostics, Inc., 6 Davis Drive, Suite 201, Research Triangle Park, NC 27709, USA; jen.miller@galaxydx.com

**Keywords:** *Babesia*, *Bartonella*, *Borrelia*, *Cytauxzoon*, Piroplasma, *Theileria*, clinical diagnostics, molecular detection, digital droplet PCR, digital PCR

## Abstract

We describe the development, optimization, and validation of a multiplex droplet digital PCR (ddPCR) assay for the simultaneous detection of *Babesia*, *Bartonella*, and *Borrelia* spp. DNA from several sample matrices, including clinical blood samples from animals and humans, vectors, in-vitro infected human and animal cell lines, and tissues obtained from animal models (infected with *Bartonella* and/or *B. burgdorferi*). The multiplex ddPCR assay was able to detect 31 *Bartonella*, 13 *Borrelia*, and 24 *Babesia* species, including *Theileria equi*, *T. cervi*, and *Cytauxzoon felis*. No amplification of *Treponema* or *Leptospira* spp. was observed. Sensitivity of 0.2–5 genome equivalent DNA copies per microliter was achieved for different members of the *Bartonella* and Borrelia genus, depending on the species or matrix type (water or spiked blood DNA) tested. The ddPCR assay facilitated the simultaneous detection of co-infections with two and three vector-borne pathogens comprising four different genera (*Babesia*, *Bartonella*, *Borrelia*, and *Theileria*) from clinical and other sample sources.

## 1. Introduction

Droplet digital PCR (ddPCR), a relatively new molecular assay, was originally developed for research and diagnostic applications in the areas of cancer and gene expression [1,2,3,4,5,6,7,8]. More recently, ddPCR technology has been adapted for enhanced molecular detection and characterization of viral, bacterial, and protozoan pathogens [9,10,11,12,13,14,15,16,17,18].

Due to characteristic ddPCR sample partitioning, this new technology provides key advantages over conventional and real-time PCR (qPCR): including increased sensitivity, unparalleled precision, and simplified absolute quantification (does not require the use of a standard curve). In addition, interference from inhibitory substances that may be present in different sample matrices is markedly reduced when using ddPCR as compared to qPCR [18]. Collectively, these features make this methodology an attractive alternative for the simultaneous amplification of vector-borne pathogen DNA, including vector-borne members of the genera *Babesia*, *Bartonella*, and *Borrelia*.

*Babesia* spp. are intraerythrocytic protozoan parasites of longstanding historical importance in human and veterinary medicine, which in recent years have been increasingly recognized as a cause of infectious disease in humans. Historically, bovine babesiosis was the very first disease proven to be tick-transmitted. Our research group was among the first to develop diagnostic PCR assays for documentation of *Babesia* spp. DNA in animal blood specimens [19,20,21,22]. Transmission of *Babesia* spp. occurs primarily by the bite of infected Ixodid ticks; however, other routes of transmission, including vertical (mother-child) and blood transfusion transmission have been reported [23,24,25]. *Babesia microti* is the causative agent for the majority of human babesiosis cases reported in the U.S., with a higher prevalence occurring in the Northeastern region of the country [26,27,28,29,30,31]. Infections caused by *B. microti* have also been reported in the upper Midwest, particularly Minnesota and Wisconsin [27,30,32,33,34], whereas published cases of *Babesia duncanii* have been reported only from the western U.S [13,35,36,37]. Human infection with *B. microti* is often asymptomatic, making this infection of particular importance in blood donor screening. The prevalence of *B. microti* within donated human blood has only recently been documented due to the historical lack of sensitive DNA-based *B. microti* tests for blood screening. Following tick transmission, babesiosis patients may display flu-like symptoms, including high fever, headaches, chills, and fatigue that may last for up to 9 weeks, whereas a subset of patients develop hemolytic anemia, at times accompanied by splenomegaly [23,29,38]. In immunocompromised individuals, including elderly or splenectomized individuals, severe cases of acute anemia, thrombocytopenia, organ failure, or even death have been reported [39]. In humans, the historical gold standard diagnostic test for detection of *Babesia* infection was microscopic examination of a blood smear, similar to established diagnostic protocols utilized for malaria diagnosis. More recently, serology and PCR-based assays (real-time and conventional PCR) have been used diagnostically, contributing to increased recognition of babesiosis as a more prevalent and significant tick-borne infection in animals and humans worldwide [37,40,41,42,43,44,45]. Importantly, the earliest investigated infectious disease applications for ddPCR were focused on the detection of *Babesia* and *Plasmodium* organisms [9,10,13]. Unfortunately, the sensitivity and specificity of these serological and molecular diagnostic assays vary among reporting laboratories, leading to confusion and controversy for both providers and patients.

Lyme disease, the most common vector-borne disease (VBD) in the USA, accounts for almost 75% of reported VBD, and affects over 300,000 people a year [46]. The disease is caused by spirochetal bacteria within the genus *Borrelia*, of which *B. burgdorferi* is the most recognized pathogenic species in the northern hemisphere. Diagnosis of acute Lyme disease is based upon symptoms (e.g., headache, fatigue, malaise, muscle pain), clinical signs (presence of an erythema migrans (EM) skin rash), and two-tier serological diagnostic testing (a positive or equivocal first tier IgM or IgG enzyme immunoassay (EIA) or immunofluorescent assay (IFA) result is confirmed by a positive second tier or reflex Western blot test) [47]. People treated with appropriate antibiotics (i.e., doxycycline, amoxicillin, or cefuroxime axetil) in the early stages of Lyme disease usually recover rapidly and completely. However, if early therapeutic intervention is not obtained (chronic Lyme disease) or if treatment fails (post-treatment Lyme disease syndrome (PTLDS)) the infection can spread to the nervous system, joints, and heart. Chronic Lyme disease and PTLDS are commonly associated with persistent symptoms, such as musculoskeletal problems; fatigue; cardiac presentations; cognitive dysfunction; headaches; sleep disturbance; and neurological presentations including demyelinating disease, peripheral neuropathy, neurodegenerative disease, and neuropsychiatric illnesses [48,49]. In addition, concurrent infection with other vector-borne pathogens in patients diagnosed with Lyme disease, including those belonging to the genera *Babesia* and *Bartonella*, further complicates and confounds clinical diagnoses and treatment approaches for the illnesses caused by this diverse group of pathogens [37,50,51,52,53,54,55].

In recent years, bartonellosis has been recognized as an emerging/re-emerging zoonotic infectious disease caused by numerous mammalian reservoir-adapted *Bartonella* species, with at least 18 *Bartonella* spp. implicated as causative agents of disease in animals or humans [56,57,58,59]. *Bartonella* species are slow growing, fastidious, facultative Gram-negative intracellular bacteria that infect a variety of mammalian hosts including companion animals, production animals, wildlife, and humans via arthropod vectors, animal bites, blood transfusion, or organ transplantation. Among others, bartonellosis is associated with a wide variety of human pathologies including endocarditis [60,61,62,63,64,65,66,67,68,69,70], cat scratch disease (CSD) [71,72,73,74,75], bacillary angiomatosis (BA) and bacillary peliosis (BP) [76,77,78,79], and neurological dysfunctions [80,81,82,83,84,85,86]. Methods of diagnosis include serological immunofluorescence assays (IFA), polymerase chain reaction (PCR), and blood cultures. However, due to their fastidious nature, complex growth requirements, cyclical, relapsing low bacteremia, and their ability to invade several cells types to subvert/evade the immune system (often leading to long delays in seroconversion and negative serology test results) [87,88,89,90,91,92,93,94,95,96,97,98], specialized diagnostic modalities, including a recently described *Bartonella* droplet digital PCR detection assay, are critically needed to improve diagnostic sensitivity [17,18,99].

We describe the development of a multiplex droplet digital PCR assay for the simultaneous detection of *Babesia*, *Bartonella*, and *Borrelia* species (BBB ddPCR) using the Bio-Rad QX One Droplet Digital PCR system. The QX ONE Droplet Digital PCR System integrates droplet generation, thermal cycling, droplet reading, and analysis into a single automated precision platform. The QX One improves upon the QX200 by providing an automated, high throughput testing platform, capable of simultaneous amplification of up to eight DNA/RNA targets with the same superior accuracy and absolute quantification achievable with qPCR. Several sample matrices were tested, including experimentally infected human and animal cell lines, spiked blood samples, animal (both domestic and wildlife) and human pre-characterized clinical samples (blood and tissue), and naturally infected sand-fly and tick species.

## 2. Methods

### 2.1. Sample Reference Types and DNA Extraction

DNA from previously characterized or as yet to be characterized *Babesia* (Piroplasma), *Bartonella*, and *Borrelia* spp., including positive and negative research and diagnostic samples from various host animals and humans submitted to the Vector-Borne Diseases Diagnostic Laboratory (VBDDL) and the Intracellular Pathogens Research Laboratory (IPRL), both at the College of Veterinary Medicine, North Carolina State University, were used to test the specificity of this BBB ddPCR assay. In addition to clinical human and animal blood and tissue samples, in vitro canine histiocytic and human epithelial experimentally infected cell lines, experimentally infected mouse tissues, naturally infected vectors (including *Ixodes* spp. ticks and sandflies), and spiked naïve human and dog blood DNA samples were tested to assess specificity and sensitivity of the assay. A descriptive list of host species and sample types are detailed for each pathogen group (Table 1: *Babesia*, *Theileria*, and *Cytauxzoon*; Table 2: *Bartonella* spp.; Table 3: *Borrelia* spp.; Table 4: Co-infections). Although our primary focus was ddPCR amplification of *Babesia* (Piroplasma), *Bartonella*, and *Borrelia* spp., the Piroplasma probe was designed to also amplify *Theileria* and *Cytauxzoon* spp., which are of veterinary medical importance.

### 2.2. Mammalian Cell Line In Vitro Infection

The ddPCR detection of intracellular *Bartonella* spp. or *Borrelia* spp. was performed on DNA extracted from experimentally infected canine histiocytic (DH82) and human epithelial (MCF10A) cell-lines cultures. Briefly, confluent DH82 (kindly provided by Henry Marr, College of Veterinary Medicine, North Carolina State University) and MCF10A (kindly provided by Dr. David Alcorta, Department of Pharmacology and Cancer Biology, Duke University, Durham, North Carolina, USA) cells were infected in vitro with either *Bartonella henselae* San Antonio 2 (strain *B. henselae* SA2) or *B. burgdorferi* clonal strain B31-MI 16 at a multiplicity of infection (MOI) of 10:1 and incubated at 37 °C with 5% CO_2_ for 24–72 h. Prior to DNA extraction, cells were subjected to a gentamicin elimination assay to eliminate extracellular bacteria [100]. Briefly, cells were treated with 150 µg/mL gentamicin for 4 h at 37 °C with 5% CO_2_ and gently washed three times with sterile PBS. Cells were harvested from each well via pipetting up and down following the addition of 1 mL of ice-cold PBS-5mM EDTA and 15-min incubation at 4 C. Cellular DNA was extracted, as described below, at 24, 48, and 72 h post-infection.

### 2.3. Experimentally Infected Mouse, Rabbit and Hamster Tissue

DNA of skin, heart, and ear tissues from experimentally and uninfected infected mice [101], kindly provided by Dr. Spector research group (Department of Pharmacology and Cancer Biology, Duke University, Durham, NC, USA), were used to assess the efficiency of *Borrelia burgdorferi* B31 detection. Similarly, blood samples collected from experimentally infected rodents, kindly provided by Dr. Sam Telford from the Department of Infectious Disease and Global Health, Tufts University, were used to assess the detection of *Babesia divergens* (rabbits), *B. duncanii* (hamsters), and *B. microti* (hamsters).

### 2.4. Spiked Blood Samples

Naïve human and dog blood was spiked with different *Bartonella* (Table 2) and *Borrelia* species (Table 3) at a final concentration of 5 genome copies per microliter for each studied pathogen. To assess the simultaneous detection of co-infections, *Babesia gibsonii* naturally infected dog blood samples were spiked with *Bartonella* spp. (*B. henselae* SA2, *B. quintana*, *B. vinsonii* subsp. *berkhoffii TI*) and/or *Borrelia burgdorferi* B31 DNA (Table 4). Similarly, naïve human blood samples spiked with *B. henselae* SA2, *B. burgdorferi* B31, and/or with *B. microti* DNA, were used to assess the detection of any combination of the three pathogens (Table 4).

### 2.5. Previously Characterized Borrelia spp. DNA Samples

A total of 94 samples, kindly provided by Drs. Volker Fingerle and Reinhard Straubinger (Laboratory Medicine, Region Jönköping County, Jönköping, Sweden) containing DNA extracted from 11 cultured *Borrelia* species at concentrations ranging from 0.1 to 2000 genome equivalents per microliter of sample (Table 5) were tested to assess ddPCR assay specificity and sensitivity. This panel was previously used for an analytical comparison of real-time PCR protocols from five different Scandinavian laboratories [102]. The ddPCR operator was blinded to the identity of all samples. Tested samples included two strains of *B. burgdorferi* (B31 and PBre); two strains of *B. afzelii* (PKo and PVPM); and five strains of *B. garinii* (PBr, PHei, P WudII, Pref and PLa), *B. spielmanii* strain PSigII, *B. bavariensis* strain PBi, *B. bissetii* strain PGeb, *B. lusitaniae* strain Poti B2, *B. valaisiana* strain VS116, *B. hermsii*, *B. miyamotoi*, and *B. turcica*. This panel also included three specificity controls, the related spirochetes *Treponema phagedenis*, and two Leptospira *interrogans* s.l. strains, both in a final concentration of 2000 genome equivalents per μL [102].

### 2.6. DNA Extraction

DNA extraction from all sample types was performed using either the Qiagen DNeasy Kit (for tissue DNA extraction) or QIAsymphony^®^ SP robot (QIAGEN, Valencia, CA, USA) and QIAsymphony^®^ DNA Mini Kit (for blood and blood culture DNA extractions), per the manufacturer’s recommended protocols. DNA quality and concentration for blood, tissue, and spiked blood samples was assessed by 260/280 nm OD measurement, as described previously [18].

## 3. Molecular Methods

### 3.1. Housekeeping Gene DNA

Housekeeping gene detection was based on either the hydroxymethyl-bilane synthase (HMBS) gene (for human host DNA) or the B-Raf Proto-Oncogene (BRAF) gene (for animal host DNA) as described previously [18], with minor modifications aimed at their detection using channel 4 of the QX One Droplet Digital PCR system (CY5.5 fluorescent dye). Briefly, housekeeping gene detection for human DNA was performed using oligonucleotides Hs-HMBS-90s (5′-TTCCTTCCCTGAAGGGATTCACTCAG-3′) and Hs-HMBS-296as (5′-TTAAGCCCAGCAGCCTATCTGACACCC-3′) as forward and reverse primers, respectively. Oligonucleotide Hs-HMBS150 (5′-CY5.5-GAAAAGCCTGTTTACCAAGGAGCTTGAACATG-BHQ_3-3′) was used as the fluorescent probe. Similarly, for detection of animal host DNA, oligonucleotides CaFeBRAF-15s (5′-TCAYGAAGACCTCACAGTAAAAATAGGT-3′) and CaFeBRAF-110as (5′-GATCCAGACAACTGTTCAAACTGATG-3′) were used as forward and reverse primers, and oligonucleotide CaFeBRAF-50 (5′-Cy5.5-GTCTAGCCACAGTGAAATCTCGATG-BHQ_3-3′) was used as the fluorescent probe.

### 3.2. Primers and Probes for Bartonella, Borrelia, and Babesia DNA Detection

*Bartonella* DNA detection, targeting a 90–120 bp segment of the intergenic spacer region (ITS) located between the 16SrRNA–23SrRNA, was performed as described previously [18]. Briefly, oligonucleotides BsppITS325s and BsppITS543as were used as sense and antisense primers respectively, and oligonucleotide BsppIT500 was used as the fluorescent probe for detection of *Bartonella* DNA amplification within channel 1 of the QX One Droplet Digital PCR system (Table 6).

*Borrelia* DNA detection, targeting a 104 bp segment of the intergenic region (ITS), was performed using oligonucleotides BobuITS120s and BoLymeITS200as, as forward and reverse primers, respectively (Table 6). Oligonucleotide BobuITS160 was used as the fluorescent probe for *Borrelia* detection within channel 2.

*Babesia* DNA detection, targeting a 125–138 bp segment (depending on species) of the 18SrRNA gene, was performed using oligonucleotides Piro18S-238s and Piro18S-380as as forward and reverse primers, respectively (Table 6). Oligonucleotide Piro18S-340 was used as the fluorescent probe for *Babesia* (Piroplasma) DNA detection within channel 3.

### 3.3. DNA Amplification

The ddPCR reaction consisted of a 20 µL final PCR reaction consisting of: 5 µL of ddPCR™ Multiplex Supermix (Bio-Rad, Hercules, CA, USA); 0.2 µL of 100 µM of each Taqman probe, forward, and reverse primer, (IDT^®^ DNA Technology, Coralville, IA, USA); 7.5 µL of molecular–grade water; 5 µL of sample DNA; 0.27 µL of 300 mM Dithiothreitol (DTT); and 1 µL of HindIII DNA restriction enzyme. The ddPCR analysis was performed using a QX One Droplet Digital PCR (Bio-Rad, Hercules, CA, USA) system under the following amplification conditions: a single hot-start cycle at 95 °C for 10 min followed by 40 cycles of denaturing at 94 °C for 30 s and annealing at 62.9 °C for 1 min. A final extension at 98 °C was performed for 5 min. Fluorescent droplet detection and distribution readings were recorded in channel 1 (for *Bartonella* DNA), channel 2 (for *Borrelia* DNA), channel 3 (for *Babesia* DNA), and channel 4 (for housekeeping DNA). Naïve blood and tissue DNA from dogs, cats, humans, and vectors (sand-flies, ticks) were used as negative controls and were assayed alongside the positive control samples within each run.

## 4. Results

### 4.1. Naive Blood, Tissue and In Vitro Cultivated Samples

Negative controls tested to assess assay specificity included over 250 pre-characterized uninfected mammalian cell culture (DH82 nor MCF10A cells) samples, uninfected human and dog blood and blood culture samples, and animal tissues. *Bartonella* (channel 1), *Borrelia* (channel 2), or *Babesia* (channel 3) DNA was not amplified from any sample. DNA amplification of the corresponding housekeeping gene (HMBS for humans, BRAF for animals) within channel 4 occurred in all uninfected-host samples (naïve cell-lines, clinical blood samples, or tissues from negative control animals). Housekeeping DNA was not amplified from the DNA extraction mixture or the negative water control (data not shown).

### 4.2. Detection of Babesia, Theileria and Cytauxzoon Species

#### Clinical Samples from Naturally and Experimentally Infected Animals

When testing pre-characterized *Piroplasma* positive DNA within animal samples (blood from naturally and experimentally infected animals, Table 1), all 23 *Babesia* species (including nine as yet to be characterized species) [20], *T. equi*, *T. cervi*, and *C. felis* spp. DNA were amplified (Table 7). Examples of reference signals of *Babesia* species DNA amplification (Channel 3) in naturally infected blood samples from dogs and other animals are represented in Figure 1.

### 4.3. Detection of Bartonella spp.

Similar to the previously reported detection using the Bio-Rad QX200 system [18], *Bartonella* DNA was amplified from a wide variety of sample matrices when the BBB ddPCR assay was conducted on the Bio-Rad QX One Droplet Digital PCR system. *Bartonella* DNA was detected within samples from naturally infected mammals (Figure 2) and from experimentally infected cell lines (dog histiocytic DH82 and human epithelial MCF10A cells), spiked naïve human and dog blood samples, and from sand-fly and tick vector tissue (results not shown). A total of 31 *Bartonella* spp. from pre-characterized human and animal clinical cases [18,103,104,105,106,107,108,109,110,111,112,113,114,115,116] were amplified using the QX-one BBB ddPCR assay described in this study (Table 8).

### 4.4. Detection of Borrelia spp.

A total of 13 *Borrelia* species (Table 9) were amplified by the BBB ddPCR assay. Species and strains detected included three Relapsing Tick-Borne Fever species (*B. turicatae*, *B. miyamotoi*, and *B. hermsii*) [117,118,119]; nine *B. burgdorferi sensu lato*-Lyme disease complex species: *B. afzelii*, *B. bavariensis*, *B. bissettii*, *B. burgdorferi ss.* (two strains), *B. coriaceae*, *B. garinii* (five strains), *B. lusitaniae*, *B. spielmanii*, *B. valaisiana* [120,121,122]; and *B. turcica*, a member of the *Borrelia* “reptile group” [123].

### 4.5. Spiked Naïve DNA Blood Samples

*Bartonella* DNA from nine species: *B. henselae* HI; *B. henselae* SA2; *B. koehlerae*; *B. melophagi*; *B. quintana*; *B. tamiae*; and *B. v. berkhoffii* genotypes I, II, and III (Table 1) was amplified when spiked into naïve dog or human blood, and they were compatible with the one found previously reported [18]. Amplification detection limits ranged between 0.5 genome equivalents per microliter for *B. henselae* HI, *B. henselae* SA2, *B. koehlerae*, *B. quintana*, and *B. v. berkhoffii* genotypes III, and 1 genome equivalent per microliterfor *B. melophagi*, *B. tamiae*, and *B. v. berkhoffii* genotypes I and II (results not shown).

Similarly, *Borrelia* DNA from seven species (Table 1) was amplified when spiked into naïve dog or human blood (Figure 3). Amplification was not observed when naïve blood DNA or water were used as templates (results not shown). The level of detection was dependent upon the *Borrelia* sp. tested. DNA from *B. burgdorferi* B31 and *B. hermsii* were detectable at 0.5 genome equivalents per microliter, whereas 5 genome copies per microliter was the lowest detectable level for *B. coriaceae*, *B. turicatae*, *B. garinii*, *B afzelii*, and *B. lusitaniae*. The level of detection for *B. bissetii* was not assessed within spiked blood, as isolates from this species were unavailable.

### 4.6. Previously Characterized Borrelia spp. DNA Samples

DNA from 12 *Borrelia* species previously used in a multi-laboratory validation panel (Table 5) that was designed to comparatively assess analytical performance of real-time PCR protocols from five different Scandinavian laboratories [102] was amplified using the BBB ddPCR assay. Detection limits of DNA spiked in water varied from 0.2 to 2 genome copy equivalents per microliter of sample (Table 10) depending on the species and strain tested. No *Borrelia* DNA was detected in negative control samples (blood samples, tissues, cell-lines, or water) nor when other non-*Borrelia* spirochetal DNA, such as from *T. phagedenis* or *Leptospira* at concentrations of 2000 spirochetes per microliter [102], were assayed at the same time, under the same conditions.

### 4.7. Tissues from Experimentally Infected Mice and Cell Lines and from Naturally Infected Vectors

*Borrelia* spp. DNA was amplified from skin, heart, and ear tissues obtained from *B. burgdorferi* B31 experimentally infected mice [101] but not from tissues samples from negative control animals. Similarly, *Borrelia* spp. DNA was amplified within intracellular fractions of experimentally infected human (MCF10A) and dog (DH82) cell lines but not within uninfected cells processed in an identical manner, and at the same time points (results not shown).

*Borrelia* spp. DNA was also amplified from naturally infected North Carolina ticks previously characterized in a separate study from our laboratory [124]. Ticks from that study assayed herein included a single *I. scapularis* tick infected with *B. burgdorferi* and single *I. affinis* ticks infected with *B. burgdorferi*, *B. bissettii*, or both *Borrelia* species [124]) (Figure 3b).

### 4.8. Dual Detection of Bartonella and Borrelia spp.

Simultaneous amplification of *Borrelia* and *Bartonella* DNA was observed for all species (*B. burgdorferi* B31, *B. henselae*, *B. quintana* and *B.v. berkhoffii* genotype II) in each respective channel when the DNA of each target organism was spiked into naive human blood samples at a concentration of 50 genome copies per microliter (Figure 4). Simultaneous detection of both *Bartonella* and *Borrelia* spp. DNA was also achieved with concentrations down to 0.01 pg/µL for each organism (equivalent to 5 bacteria genome copies per microliter of blood). Although the limit of detection for *Bartonella* DNA within dual spiked samples was 0.5 genome copies per microliter, as previously reported for spiked samples containing only *Bartonella* DNA [125], amplification of *Borrelia* DNA was not achievable within dual spiked samples at 0.5 genome copies per microliter (results not shown). Simultaneous *Borrelia* and *Bartonella* DNA amplification was observed within previously characterized naturally co-infected ticks [124,126,127] (results not shown).

### 4.9. Dual Detection of Bartonella and Babesia

The BBB ddPCR assay was able to detect natural, dual infection with *B. vinsonii* and *Babesia vulpes* within blood samples obtained from gray foxes from Portugal (Figure 5). Consistent mean fluorescent intensities and droplet distribution patterns were obtained for each pathogen within their respective channels.

### 4.10. Babesia, Bartonella, and Borrelia Spiked Naïve Blood Samples

DNA from all three species, *Babesia* (*Babesia microti*), *Bartonella* (*B. henselae*), and *Borrelia* (*Borrelia burgdorferi* B31), were detected when spiked into DNA extracted from naïve human blood (Figure 6) or when *Bartonella* (*B. henselae*) and *Borrelia* (*Borrelia burgdorferi* B31) DNA were spiked (at concentrations of 5 genome equivalent copies per microliter) into DNA extracted from a naturally *B. gibsonii* infected dog blood (results are not shown).

## 5. Discussion and Conclusions

We developed and optimized a multiplex droplet digital PCR assay, using the Bio-Rad QX One System that can simultaneously amplify DNA from *Babesia*, *Bartonella*, and *Borrelia* species. The assay (BBB ddPCR) amplified DNA from 31 *Bartonella* species, 13 *Borrelia* species (from the Lyme, relapsing fever, and reptile complex), and 24 *Babesia* species. The assay also amplified two *Theileria* spp. (*T. equi* and *T. cervi*), as well as *C. felis* DNA from naturally infected clinical animal blood specimens. The multiplex BBB ddPCR assay presented herein reliably detected single and co-infections involving vector-borne pathogens from the genera *Babesia*, *Bartonella*, *Borrelia*, and *Theilaria*, using a variety of animal and human clinical samples, vectors, and experimentally infected tissues and cell-lines. The assay did not amplify *Babesia*, *Bartonella*, or *Borrelia* species DNA (no droplets observed) in multiple negative control samples (tissues, naïve cell-lines, naïve and clinical blood specimens, or water), tested at the same time and under the same conditions. The ability to co-amplify multiple vector-borne pathogens within a single sample with high sensitivity will greatly enhance the efficiency and efficacy of clinical diagnostic testing, particularly of volume-limited or otherwise hard to obtain sample matrices.

Despite the high analytical specificity and low limit of detection measured for the *Bartonella* and *Borrelia* spp. tested with the BBB ddPCR assay, one limitation of the current Bio-Rad QX One System is the inability to concentrate amplified DNA for genus and species confirmation by DNA sequencing. In addition to other diagnostic and epidemiological considerations, this limitation is of particular clinical relevance for *Babesia* species in veterinary medicine, where the treatment protocol varies depending upon the infecting Piroplasma (large versus small *Babesia*) spp. Sequence-based confirmation of pathogen identity is also critical in the context of chronic, stealth, and/or low-yield infections both to fulfill Koch’s postulates, which stipulate identification/isolation of the disease-causing pathogen, as well as to ensure that the proper and most effective anti-microbial therapies are being administered to patients. This limitation is also critical for hard to obtain or volume-limited samples for which insufficient sample may remain for additional testing, or for situations where no secondary companion confirmatory test modality is available.

Compared to currently available molecular diagnostic modalities, it is anticipated that ddPCR will provide both exceptional sensitivity and specificity for the diagnosis of babesiosis, bartonellosis, and borreliosis within animal and human patients. In addition, the previously reported clinical enhanced sensitivity of ddPCR [9,13,16,18] will facilitate the discovery and subsequent characterization of novel organisms infecting animals, humans, and vectors. In contrast to serological assays, ddPCR will also enhance the capability of diagnostic laboratories to confirm a molecular diagnosis of co-infections by providing the ability to simultaneously assay multiple combinations of vector-borne pathogens and will shorten the sample to answer window for providers by reducing the number of tests to be performed on a single patient sample. Co-infections in animals and human patients induce increased clinical complexity, present more robust diagnostic challenges, and greatly influence and complicate treatment decisions. Future studies aimed at the addition of other vector-borne organisms such as *Anaplasma*, *Ehrlichia*, and *Rickettsia* species to the existing BBB ddPCR platform, without decreasing assay sensitivity, would be highly beneficial for clinical and research applications in human and veterinary medicine.

## Figures and Tables

**Figure 1 pathogens-10-01462-f001:**
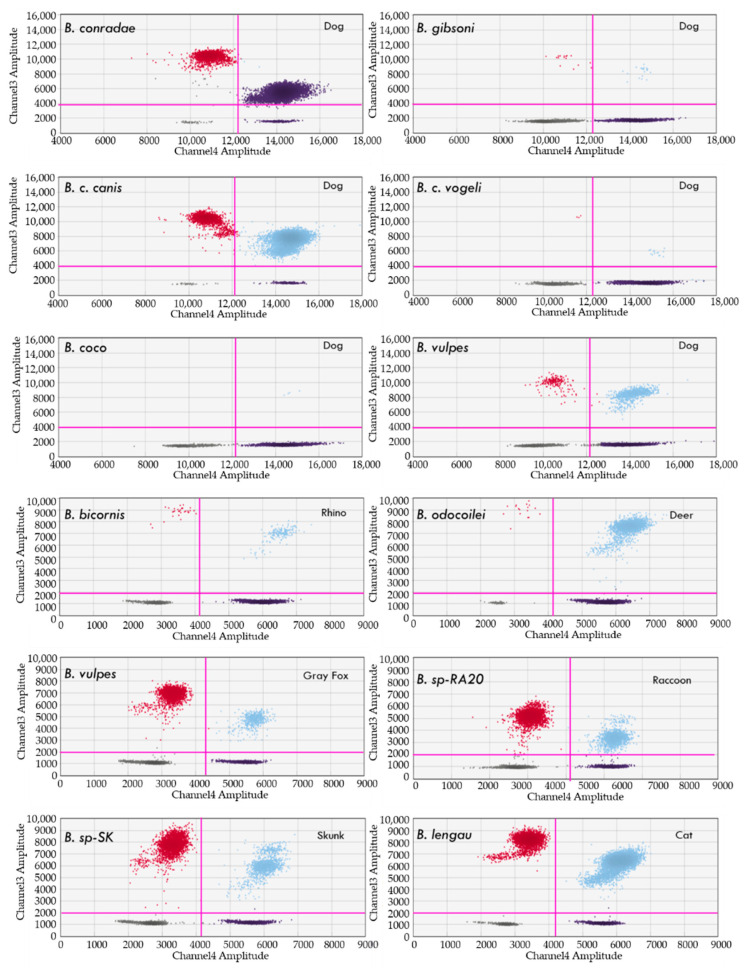
Detection of different *Babesia* species (red and light blue dots) in blood from naturally infected dogs, cats, deer, raccoons, gray foxes, skunks, and rhinoceros. Note: Only channel 3 (piroplasmas) and channel 4 (housekeeping, in dark black color) are shown in each graph. sp.: not characterized species.

**Figure 2 pathogens-10-01462-f002:**
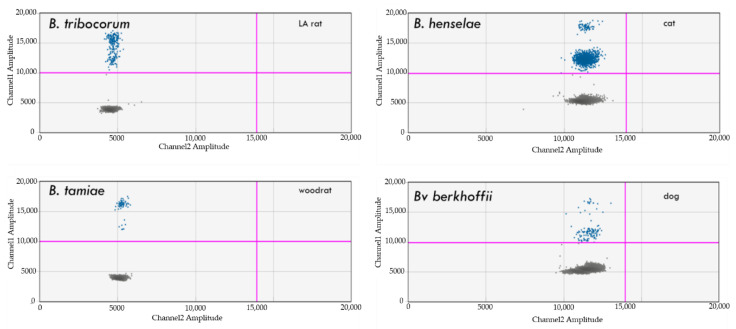
Detection of *Bartonella* spp. (blue dots) in blood samples from different animal blood samples. Note: Only channel 1 (*Bartonella*) and Channel 2 (*Borrelia*) are shown in each graph.

**Figure 3 pathogens-10-01462-f003:**
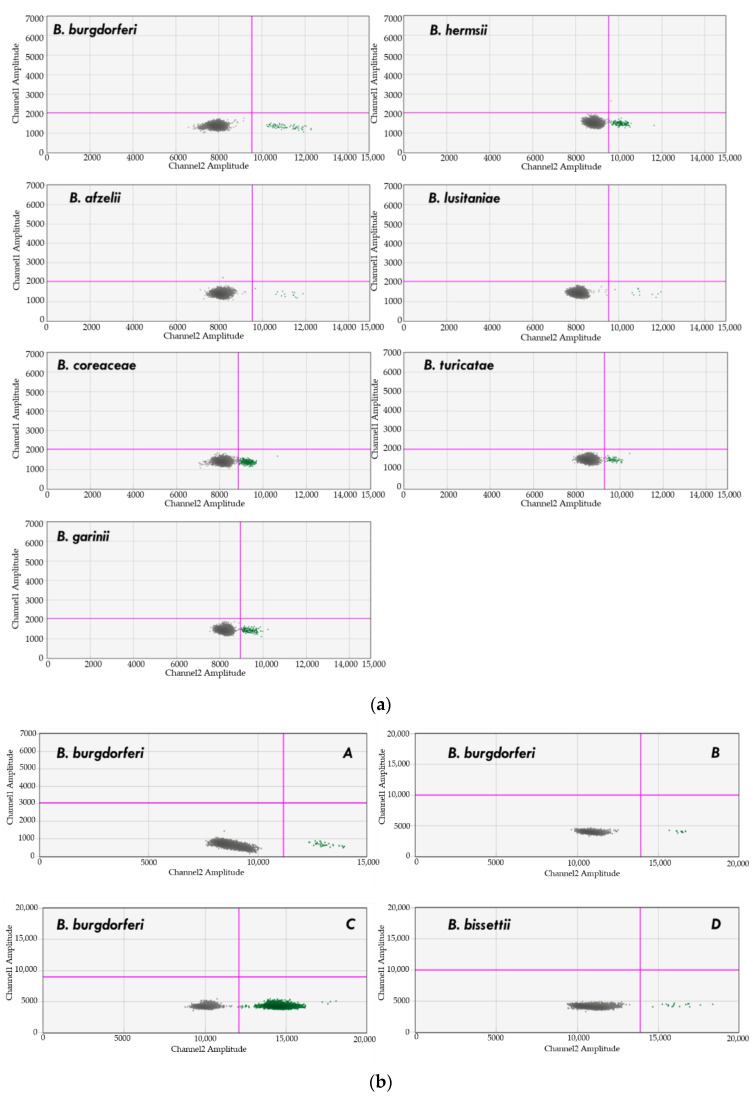
(**a**) Detection of *Borrelia burgdorferi*, *B. coreaceae*, *B. garini*, *B. hermsii*, *B. afzelii*, *B. lusitanae*, and *B. turicatae* (green dots) DNA in spiked naïve human blood. Note: Only channel 1 (*Bartonella*) and Channel 2 (*Borrelia*) are shown in each graph; (**b**) Detection of *Borrelia* spp. (green dots) DNA in: (**A**) *B. burgdorferi* B31 in vitro infected mice (skin sample); (**B**) *B. burgdorferi* ss in *I. scapularis* infected tick; (**C**) *B. burgdorferi* B31 in vitro infected MCF10A human epithelial cells; (**D**) *B. bissettii* in *I. affinis* infected tick. Note: Only channel 1 (*Bartonella*) and Channel 2 (*Borrelia*) are shown in each graph.

**Figure 4 pathogens-10-01462-f004:**
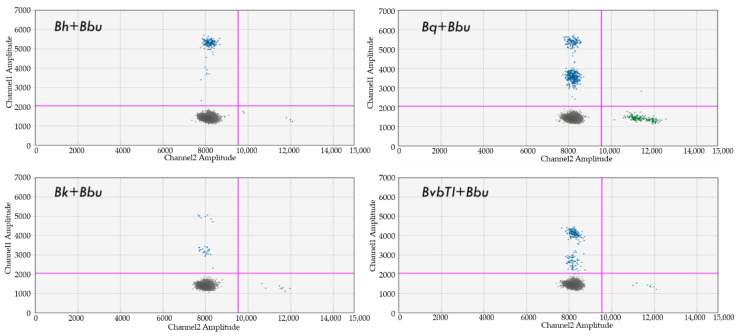
Naïve human DNA spiked with B. burgdorferi B31 (green dots) and different *Bartonella* spp. (blue dots). Note: Channel 1 (Bartonella) and Channel 2 (Borrelia) are shown only; Bh: Bartonella henselae; Bq: Bartonella quintana; Bk: Bartonella koehlerae; BvbTI: Bartonella vinsoni berkhoffii genotype I; Bbu: Borrelia burgdorferi B31.

**Figure 5 pathogens-10-01462-f005:**
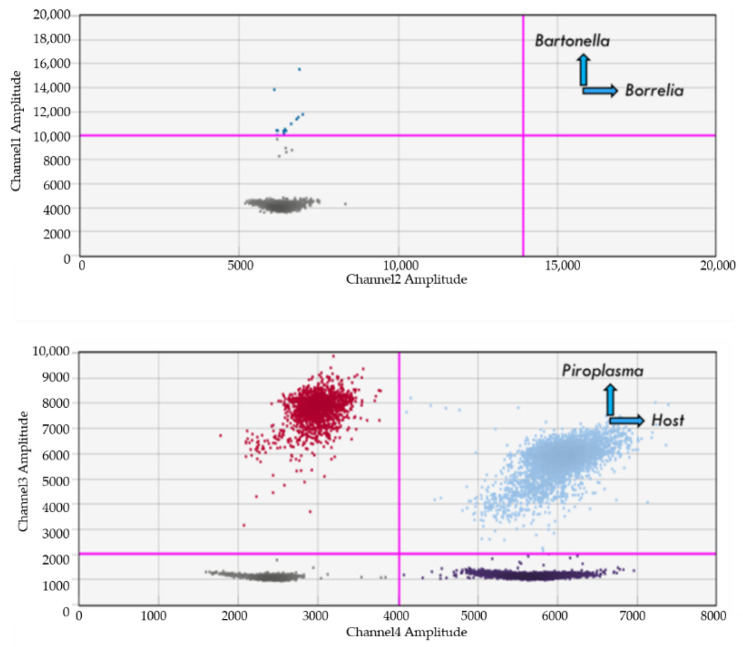
Co-detection of *B. vinsonii* berkhoffii (blue dots) and *Babesia vulpes* (red and light blue dots) DNA in a naturally infected gray fox’s blood sample. Note: dark color dots represent host housekeeping gene (HK).

**Figure 6 pathogens-10-01462-f006:**
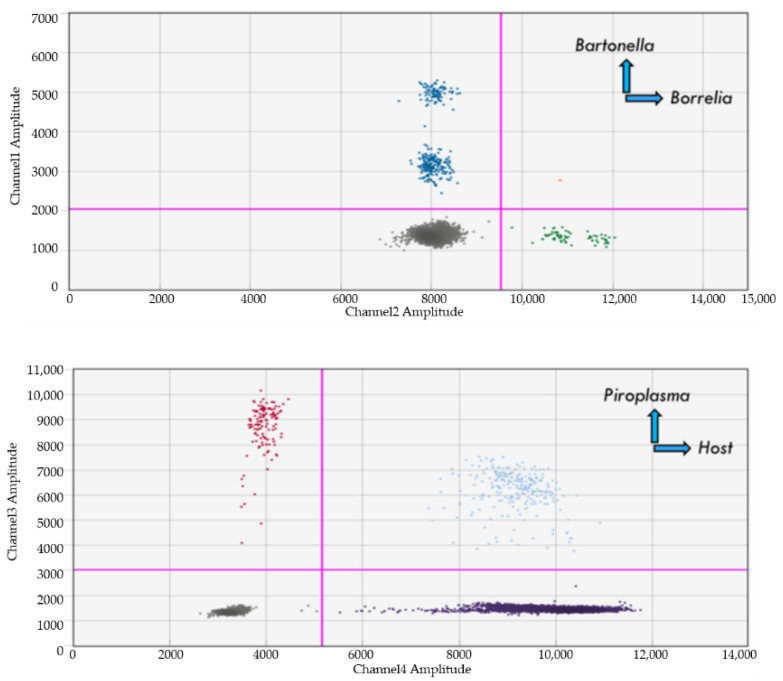
Simultaneous detection of *Babesia* (red and light blue dots), *Bartonella* (blue dots) and *Borrelia* (green dots) DNA in blood. Naïve human blood DNA spiked with *Bartonella henselae*, *Borrelia burgdorferi* B31, and *Babesia microti* DNA at 50 genome copies per microliter each. Note: dark color dots represent host housekeeping gene (HK).

**Table 1 pathogens-10-01462-t001:** List of *Piroplasma* species tested by host and sample types.

Piroplasma	Sample Type	Host/s	Infection Type
*B. bicornis*	Blood	Rhinocerous	Naturally
*B. canis canis*	Blood	Dogs	Naturally
*B. canis vogeli*	Blood	Dogs	Naturally
*B. coco*	Blood	Dogs, Horse	Naturally
*B. conrade*	Blood	Dogs	Naturally
*B. divergens*	Blood	Rabbit	Experimentally
*B. duncani*	Blood	Hamster	Experimentally
*B. felis*	Blood	Cats	Naturally
*B. gibsoni*	Blood	Dogs	Naturally
*B. gibsoni* + *B. vulpes*	Blood	Dog	Naturally
*B. lengau*	Blood	Cats	Naturally
*B. microti*	Blood	Hamster	Experimentally
*B. odoncoilei*	Blood	Deer, Reindeer	Naturally
*B. vulpes*	Blood	Dogs, Foxes	Naturally
*B. vulpes* + *Babesia* spp.	Blood	Raccoon	Naturally
*Babesia* spp.	Blood	Dog	Naturally
*Babesia* spp. *AJB 2006*	Blood	Bear	Naturally
*Babesia* spp. *NC08*	Blood	Bear	Naturally
*Babesia* spp. *NC13*	Blood	Bear	Naturally
*Babesia* spp. *R-214*	Blood	Racoon	Naturally
*Babesia* spp. *R431*	Blood	Racoon	Naturally
*Babesia* spp. *R-9879*	Blood	Racoon	Naturally
*Babesia* spp. *R-CO8*	Blood	Racoon	Naturally
*Babesia* spp. *SK04*	Blood	Skunk	Naturally
*Babesia* spp. *SK08*	Blood	Skunk	Naturally
*Babesia* spp. *WR3*	Blood	Woodrat	Naturally
*C. felis*	Blood	Cats	Naturally
*Maned wolf Babesia*	Blood	Dogs, Racoons	Naturally
*T. cervi*	Blood	Deer, Bears	Naturally
*T. equi*	Blood	Horses	Naturally

DNA source: blood, tissue, cells; Infection type: Naturally infected, experimentally infected, spiked sample; Abbreviations: spp.: denotes an uncharacterized *Babesia* species.

**Table 2 pathogens-10-01462-t002:** List of *Bartonella* species tested by host and sample types.

Bartonella	Sample Type	Host/s	Infection Type
*B. alsatica*	Blood	Rabbits	Naturally
*B. bovis*	Blood	Cows	Naturally
*B. chomelii*	Blood	Elk	Naturally
*B. clarridgeiae*	Blood	Cats, Dogs	Naturally
*B. henselae*	Blood	Humans, Cats, Dogs	Naturally
*B. henselae*	Tissue (Brain)	Dolphins	Naturally
*B. henselae*	Tissue (Body)	Fleas, Sanflies	Naturally
*B. henselae*	Tissue (Skin)	Mice	Experimentally
*B. henselae*	Tissue (Body)	Tick (*I. scapularis*)	Naturally
*B. henselae H1*	Blood	Human, Cat, Dog	Spiked
*B. henselae SA2*	Blood	Human, Dogs, Birds	Naturally
*B. henselae SA2*	Blood	Human, Dog	Spiked
*B. henselae SA2*	DH82 cell-line	Dog	Experimentally
*B. henselae SA2*	MCF10A cell-line	Human	Experimentally
*B. koehlerae*	Blood	Human, Dog	Spiked
*B. melophagi*	Blood	Human, Dog	Spiked
*B. melophagi*	Blood	Sheep	Naturally
*B. quintana*	Blood	Human, Dog	Spiked
*B. quintana*	Blood	Human, Monkey	Naturally
*B. rochalimae*	Blood	Dogs	Naturally
*B. tamiae*	Blood	Human	Spiked
*B. tamiae*	Blood	Woodrats	Naturally
*B. tribocorum*	Blood	Rats	Naturally
*B. v. berkhoffii TI*	Blood	Human, Dog	Spiked
*B. v. berkhoffii TII*	Blood	Human, Dog	Spiked
*B. v. berkhoffii TII*	Blood	Human, Coyotes, Foxes	Naturally
*B. v. berkhoffii TIII*	Blood	Human, Dog	Spiked
*B. v. berkhoffii TIII*	Blood	Skunk	Naturally
*B. volans*	Tissue (Heart)	River otter	Naturally
*Bartonella* spp.	Blood	Mongooses	Naturally
*Bartonella* spp. *Rc*	Blood	Racoons	Naturally

Sample type: blood, tissue, cells; Infection type: Naturally infected, experimentally infected, spiked sample. Abbreviations: *B. v. berkhoffii TI*: *B. vinsoni berkhoffii* genotype I; *B. v. berkhoffii TII*: *B. vinsoni berkhoffii* genotype II; *B. v. berkhoffii TIII*: *B. vinsoni berkhoffii* genotype III; spp.: denotes an uncharacterized *Bartonella* species.

**Table 3 pathogens-10-01462-t003:** List of *Borrelia* species tested by host and sample types.

Borrelia	Sample Type	Host	Infection Type
*B. afzelii*	Blood	Dog	Spiked
*B. bissetii*	Tissue (Body)	Tick (*I. affinis*)	Naturally
*B. burgdorferi*	Tissue (Body)	Tick (*I. scapularis*)	Naturally
*B. burgdorferi B31*	Blood	Human, Dog	Spiked
*B. burgdorferi B31*	DH82 cell line	Dog	Experimentally
*B. burgdorferi B31*	MCF10A cell line	Human	Experimentally
*B. burgdorferi B31*	Tissue (Skin)	Mice	Experimentally
*B. coreaceae*	Blood	Dog	Spiked
*B. garinii*	Blood	Dog	Spiked
*B. hermsii*	Blood	Dog	Spiked
*B. lusitanae*	Blood	Dog	Spiked
*B. turicatae*	Blood	Dog	Spiked

DNA source: blood, tissue, cells; Infection type: Naturally infected, experimentally infected, spiked sample.

**Table 4 pathogens-10-01462-t004:** List of co-infecting species tested by host and sample types.

Co-Infection	Sample Type	Host	Infection Type
*B. burgdorferi* + *B. gibsonii*	Blood	Dog	Naturally + Spiked
*B. burgdorferi* + *B. gibsonii*	Blood	Dog	Naturally + Spiked
*B. burgdorferi* + *B. microti*	Blood	Human	Spiked
*B. henselae* + *B. burgdorferi*	Blood	Dog	Spiked
*B. henselae* + *B. burgdorferi*	Blood	Human	Spiked
*B. henselae* + *B. burgdorferi*	Tissue (Brain)	Human	Experimentally
*B. henselae* + *B. burgdorferi*	Tissue (Body)	Tick (*I. affinis*)	Naturally
*B. henselae* + *B. burgdorferi* + *B. gibsonii*	Blood	Dog	Naturally + Spiked
*B. henselae* + *B. burgdorferi* + *B. microti*	Blood	Human	Spiked
*B. henselae* + *B. burgdorferi B31*	Tissue (Skin)	Mice	Experimentally
*B. henselae* + *B. microti*	Blood	Human	Spiked
*B. odoncoilei + B. henselae*	Tissue (Body)	Tick (*I. affinis*)	Naturally
*B. quintana* + *B. burgdorferi*	Blood	Dog	Spiked
*B. quintana* + *B. burgdorferi*	Blood	Human	Spiked
*B. v. berkhoffii TII + B. burgdorferi*	Blood	Dog	Spiked
*B. v. berkhoffii TII* + *B. burgdorferi*	Blood	Human	Spiked
*B. vinsonii* + *B. vulpes*	Blood	Gray fox	Naturally

**Table 5 pathogens-10-01462-t005:** List of DNA from 11 cultured *Borrelia* species tested to assess specificity and sensitivity [102].

*Borrelia* sp.	Strain	Conc. Range (Genome Copies per 5 µL Sample)
*B. afzelii*	Pko	0.1–10,000
*B. bavariensis*	VS116	0.1–10,000
*B. bissettii*	Pgeb	10,000
*B. burgdorferi*	B31, Pbre	0.1–10,000
*B. garinii*	PBr, Phei, Pla, Pref, PWudII	0.1–10,000
*B. hermsii*		0.1–10,000
*B. lusitanae*	PotiB2	0.1–10,000
*B. miyamotoi*		10,000
*B. spielmanii*	PSigHH	0.1–10,000
*B. turcica*		10,000
*B. valaisiana*	VS116	0.1–10,000

**Table 6 pathogens-10-01462-t006:** ddPCR primers and probes for *Babesia*, *Bartonella*, and *Borrelia* amplification and detection.

Primer/Probe	Name	Channel	Sequence
*Babesia* sense	Piro18S-238s	3	5′-TCGGTGATTCATAATAAACTRGCGAATCGC-3′
*Babesia* antisense	Piro18S-380as		5′-GAATCGAACCCCAATTCCCCGTTACCCG-3′
*Babesia* probe	Piro18S-340		5′-Cy5- GACGGTAGGGTATTGGCCTACCG-BHQ2-3′
*Bartonella* sense	BsppITS325s	1	5’-CCTCAGATGATGATCCCAAGCCTTCTGGCG-3’
*Bartonella* antisense	BsppITS543as		5′-TAAAYTGGTGGGCCTGGGAGGACTTG-3′
*Bartonella* probe	BsppITS500		5′-FAM-GTTAGAGCGCGCGCTTGATAAG -IABkFQ-3′
*Borrelia* sense	BobuITS120s	2	5′-AGGTCATTTTGGGGGTTTAGCTCAGTTGGCT-3′
*Borrelia* antisense	BoLymeITS200as		5′-AATGGAGGTTAAGGGACTCGAACCCT-3′
*Borrelia* probe	BobuITS160		5′-HEX-CGGCTTTGCAAGCCGAGGGTCAAG-BHQ-2-3′

**Table 7 pathogens-10-01462-t007:** *Babesia*, *Theileria*, and *Cytauxzoon* species detected using the BBB ddPCR.

*Babesia* Species		*Theileria-Cytauxzoon*
*B. bicornis*	*B. microti*	*T. cervi*
*B. bigemina*	*B. negevi-like*	*T. equi*
*B. c. canis*	*B. vulpes* (former *B. annae*)	*C. felis*
*B. c. vogeli*	maned Wolf Babesia	
*B. coco*	Bsp AJB 2006 *	
*B. conradae*	Bsp woodrat *	
*B. divergens*	Bspp skunks (SK04 *, SK08 *)	
*B. duncani*	Bspp raccoons (R214 *, 9879 *, CO8 *)	
*B. gibsoni*	Bspp bears (NC08 *, NC13 *)	
*B. lengau*		

Note: * uncharacterized species.

**Table 8 pathogens-10-01462-t008:** *Bartonella* species detected using the BBB ddPCR.

*Bartonella* Species			
*B. henselae*	*B. alsatica*	*B. doshiae*	*B. rochalimae*
*B. koehlerae*	*B. melophagi*	*B. schoembuchensis*	*B. tamiae*
*B. quintana*	*B. volans*	*B. rattimassiliensis*	*Ca*. B. k. boulousii
*B. v. berkhoffii TI*	*B. elizabethae*	*B. tribocorum*	*Ca*. B. k. bothierii
*B. v. berkhoffii TII*	*B. monaki*	*B. chomelii*	Bsp deer (D2 *, D4 *, D5 *, D6 *)
*B. v. berkhoffii TIII*	*B.v. baker*	*B. washoensis*	Bsp river otter *
*B. clarridgeiae*	*B. bovis*	*B. vinsonii*	Bsp mongoose *

* denotes uncharacterized species and their hosts.

**Table 9 pathogens-10-01462-t009:** *Borrelia* species detected by the BBB ddPCR assay.

*Borrelia* Species (Strains)	
*B. afzelii* (Pko)	*B. lusitanae* (PotB2)
*B. bavariensis* (PBi)	*B. miyamotoi*
*B. bissettii* (PGeb)	*B. spielmanii* (PSigII)
*B. burgdorferi* (B31, PBre)	*B. turcica*
*B. coriaceae*	*B. turicatae*
*B. garinii* (PBr, PHei, Pla, PRef, PWudII)	*B. valaisiana* (VS116)
*B. hermsii*	

**Table 10 pathogens-10-01462-t010:** Analytical performance of the BBB ddPCR assay for the 11 *Borrelia* species from the validation panel used in this study [102].

*Borrelia* sp.	Strain	Detection Limit (Genome Copies per 5 µL Sample)
*B. afzelii*	Pko	1–10
*B. bavariensis*	VS116	0.1–10
*B. bissettii*	Pgeb	1–10
*B. burgdorferi*	B31, Pbre	1–10
*B. garinii*	PBr, Phei, Pla, Pref, PWudII	1–10
*B. hermsii*		NA
*B. lusitanae*	PotiB2	1–10
*B. miyamotoi*		NA
*B. spielmanii*	PSigHH	1–10
*B. turcica*		1–10
*B. valaisiana*	VS116	10

NA: Not assayed.

## Data Availability

Data supporting reported results are available upon request. Please, contact rgmaggi@ncsu.edu.

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
