# Peer review of "Development of a Multiplex Droplet Digital PCR Assay for the Detection of Babesia, Bartonella, and Borrelia Species"

_pathogens, 2021, doi:10.3390/pathogens10111462_

Round 1
Reviewer 1 Report
Comments to the author
The manuscript “Development of a multiplex droplet digital PCR assay for the detection of Babesia, Bartonella, and Borrelia species” describes the development, optimization, and validation of a multiplex droplet digital PCR (ddPCR) assay for the simultaneous detection of Babesia, Bartonella, and Borrelia spp. This methodology offers a better diagnosis of babesiosis, bartonellosis, and borreliosis for animal and human patients.
The manuscript is well-presented and well-written. The experimental design is well-done because it evaluated different human and animal samples naturally or experimentally infected with different species of the target microorganisms.
I suggest that some aspects of the manuscript improve before publication. The discussion needs to address some left questions: Can this new system be used as a regular diagnosis tool? What is the cost-benefit of this new technology? Some abbreviations need to be detailed as for example HMBS and BRAF gene in the methodology section. Additionally, the authors can describe the weakness of the study and propose alternatives to address it. It is easier to understand the graphs when all figures and legend figures keep the same format, so I suggest that all figures and legend figures are unified as follow:
All legend figures should explain what represent each color; for example, fig 5 and fig 6 do not have the information about the color used. Some figures do not have labels of the identified species inside the graph as for example fig 3b. What is the meaning of the pink square in some figures? Is there a separation of host and microorganism? Fig 1 does not have the pink square. Figs 1 and 2 have outside borders but the other figures in the manuscript do not have them. All figures should have the same size and font that are readable.
Author Response
We would sincerely thank the comments and suggestions made by the reviewer. Below, are the details on the response to the reviewer’s questions and queries:
- Can this new system be used as a regular diagnosis tool?
- This technology can and it has been used as regular diagnostic tool. Some examples of current assays commercially available for the ddPCR systems from Bio-Rad are the Microsatellite instability assay; SARS-CoV-2 ddPCR test; Assays for AAV viral titer; and the Vericheck ddPCR Mycoplasma detection assay (all from Bio-Rad). Other companies offering digital PCR systems, either as droplet digital PCR or plate-based digital PCR include 4basebio AG; Fluidigm Corporation; Combinati, Inc.; JN Medsys; Merck KGaA; Qiagen N.V.; Stilla Technologies; Sysmex Corporation; and Thermo Fisher Scientific. In the arena of vector-borne pathogens Galaxy Diagnostics is an example of a commercial diagnostic company applying this technology for Bartonella detection. Moreover, digital PCR technology has been increasingly used at the point of care clinical settings. These POC systems are mostly based on microfluidic array plate (MAP) technology and can provide with similar characteristics of absolute quantification of low abundant targets as found for droplet digital PCR (i.e. FastPlex Triplex for SARS-CoV-2 Detection, or the somatic mutations using liquid biopsy dPCR assays).
- Some of these aspects can be included in the discussion if the reviewer and/or editor consider it worth to mention.
- What is the cost-benefit of this new technology?
- The two most important cost-benefits of this technology are the capability of multiplexing while providing precise and absolute quantification of target genes, and its higher sensitivity (as compared with qPCR), allowing detection of nucleic acids (i.e. testing allelic variants or low abundance pathogens) that are below the sensitivity limit of qPCR. Interestingly, the cost per run using ddPCR are similar to the one used in “standard” real-time qPCR assays. Currently the equipment is expensive, as is the yearly maintenance contract. So startup costs for ddPCR exceed startup costs for qPCR testing.
- Some abbreviations need to be detailed as for example HMBS and BRAF gene in the methodology section.
- Clarification on the identification of each gene were included in the methods section
- The authors can describe the weakness of the study and propose alternatives to address it.
- Despite the high analytical specificity and low limit of detection that can be achieved by ddPCR, one limitation of this and similar systems when applied to vector-borne pathogens, is the inability to identify the species detected in a sample (if, as in out assay, the test is aimed at detecting pathogens at the genus level). In this particular case, additional tests should be performed. Two of the most common alternatives will be qPCR using the same primers and probes to be able to identify the pathogen at species and strain level by DNA sequencing, or to develop ddPCR species-specific probes. As described in the text, the limitation of pathogen ID when a genus detection test is performed is of particular clinical relevance for some vector-borne pathogens groups, such as for Babesia species. Babesia species treatments, in the veterinarian clinical setting vary depending upon the infecting Piroplasma (large versus small Babesia) species.
To solve this problem, our laboratory is currently working on two fronts: a) developing an additional species-specific ddPCR assay and b) developing new/improved methodologies to capture higher quantities of DNA from clinical samples that will allow secondary tests (such as qPCR) aimed at DNA sequencing.
- It is easier to understand the graphs when all figures and legend figures keep the same format, so I suggest that all figures and legend figures are unified as follow: All legend figures should explain what represent each color; for example, fig 5 and fig 6 do not have the information about the color used.
- The clarification of what detection represents each drop color has been added.
- Some figures do not have labels of the identified species inside the graph as for example fig 3b. What is the meaning of the pink square in some figures? Is there a separation of host and microorganism? Fig 1 does not have the pink square. Figs 1 and 2 have outside borders but the other figures in the manuscript do not have them. All figures should have the same size and font that are readable.
- The clarification of what is represented in each graph has been added. The pink lines represent the threshold detection lines. These have been included in all graphs where they were missing
- The fonts had been changed to match uniformly throughout.
Reviewer 2 Report
A multiplex droplet digital PCR has more advantages in screening of several pathogens compared to conventional PCR and real-time PCR. The ddPCR developed should be useful tool for molecular epidemiology of the pathogens from now on. Thus, the method may help colleagues in this research filed to make new study design.
There is no concern regarding the study contents and the conclusions, but I'm a little worried about the number of references in the study. I recommend that you are better to select only the references needed for the study's background and your results.
Author Response
We would sincerely thank the comments and suggestions made by the reviewer. Below, are the details on the response to the reviewer’s questions and queries:
- There is no concern regarding the study contents and the conclusions, but I'm a little worried about the number of references in the study. I recommend that you are better to select only the references needed for the study's background and your results.
- We understand the concern pointed by the reviewer. Nevertheless, we already tried to limit the number of references made in the manuscript to the minimum we considered important to be mentioned. If the reviewer and Editor still consider them to be too large, the authors may try again to limit its number even further, but we kindly request all of them to be included if possible.
Reviewer 3 Report
Comments to the manuscript entitled "Development of a multiplex droplet digital PCR assay for the 2 detection of Babesia, Bartonella, and Borrelia species" based on a large material collected. This paper is an original, scientifically solid and important contribution to the study of parasites and other vector-borne pathogens reporting molecular identification by new methods multiplex droplet digital PCR assay.
Comments:
Table 1a: it is well known, but for completeness it would be useful to explain the abbreviations C. (Cytauxzoon) and T. (Theileria) in the table.
Line 169: Experimentally infected mouse, rabbit and hamster tissue – please indicate who authorized the experimental infection and under what number.
Line 211-212: Could you justify why you chose these genes HMBS – human host; BRAF – animal host?
Describe in more detail what it is minor modification – line 212-213.
Author Response
We would sincerely thank the comments and suggestions made by the reviewer. Below, are the details on the response to the reviewer’s questions and queries:
- Table 1a: it is well known, but for completeness it would be useful to explain the abbreviations C. (Cytauxzoon) and T. (Theileria) in the table.
- The full genus names was included in each row of the table
- Line 169: Experimentally infected mouse, rabbit and hamster tissue – please indicate who authorized the experimental infection and under what number.
- The experimental mouse tissue samples were provided by Duke Cancer Center during a collaborative work with Tulane University. The samples were part of DOI: https://doi.org/10.1128/JCM.02313-20. Practices in the housing and care of mice conformed to the regulations and standards of the Public Health Service Policy on Humane Care and Use of Laboratory Animals, and the Guide for the Care and Use of Laboratory Animals. The Tulane National Primate Research Center (TNPRC) is fully accredited by the Association for the Assessment and Accreditation of Laboratory Animal Care-International (Animal Welfare Assurance A4499-01). The Tulane University Institutional Animal Care and Use Committee approved all animal-related protocols.
- The experimental rabbit and hamster blood samples were provided by Dr. Sam Telford from the Department of Infectious Disease and Global Health, Tufts University under IACUC G2019-32 and G2020-104.
- Line 211-212: Could you justify why you chose these genes HMBS – human host; BRAF – animal host?
- We choose the HMBS and the BRAF genes as human and animal reference genes since both of them were used in our original work for the development of the Bartonella droplet digital PCR and other previously published works:
- Maggi, R et al. “Development and validation of a droplet digital PCR assay for the detection and quantification of Bartonella species within human clinical samples”, J Microbiol Methods 2020; 176:106022;
- Lashnits, E et al. “Comparison of Serological and Molecular Assays for Bartonella Species in Dogs with Hemangiosarcoma”, Pathogens 2021; 10(7):794. doi: 10.3390.
- Lashnits, E et al. “Schizophrenia and Bartonella spp. Infection: A Pilot Case-Control Study” Vector Borne Zoonotic Dis. 2021 21(6):413-421. doi: 10.1089/vbz.2020.2729
- The selection of these two housekeeping (HK) genes for this work was to maintain consistency in the assessment of the host amplification quality control. Both genes have been used as reference in several other works as well: The hydroxymethyl-bilane synthase (HMBS) was found to be the single best reference gene for gene expression studies (optimal reference gene for normalizing gene expression data) in a work done by Cicinnati and other in 2008 (PMID: 19036168; DOI: 10.1186/1471-2407-8-350). Similarly, the BRAF (B-Raf Proto-Oncogene, Serine/Threonine Kinase) gene was used in this work as reference since it has been used previously for assessing mutation in urine DNA as a molecular diagnostic for canine urothelial and prostatic carcinoma by ddPCR (Christensen et al. Identification of robust reference genes for studies of gene expression in FFPE melanoma samples and melanoma cell lines. Melanoma Res. 2020 30(1):26-38)
- We choose the HMBS and the BRAF genes as human and animal reference genes since both of them were used in our original work for the development of the Bartonella droplet digital PCR and other previously published works:
- Describe in more detail what it is minor modification – line 212-213.
- Minor modifications are referred to changes made in the fluorescent dye and quenchers used in the housekeeping genes probes. In the previous studies (see references above) the dyes and quencher for the HK genes probes were HEX and BHQ-2 respectively since only two channels were available for testing using the Bio-Rad QX 200 system. The QXOne system used in this work, allows four channels for fluorescence detection. The HK detection was allocated in channel 4 for this work, where Cy5.5 can be used as a fluorescent dye. The changes were, then switching Hex/BHQ-2 for Cy5.5/BHQ2 as dye/quencher for each HK probe.